# Post-Newtonian Jeans Equation for Stationary and Spherically Symmetrical Self-Gravitating Systems

Gilberto Medeiros Kremer 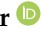

Departamento de Física, Universidade Federal do Paraná, Curitiba 81531-980, Brazil; kremer@fisica.ufpr.br

**Abstract:** The post-Newtonian Jeans equation for stationary self-gravitating systems is derived from the post-Newtonian Boltzmann equation in spherical coordinates. The Jeans equation is coupled with the three Poisson equations from the post-Newtonian theory. The Poisson equations are functions of the energy-momentum tensor components which are determined from the post-Newtonian Maxwell–Jüttner distribution function. As an application, the effect of a central massive black hole on the velocity dispersion profile of the host galaxy is investigated and the influence of the post-Newtonian corrections are determined.

**Keywords:** post-Newtonian theory; Boltzmann equation; Jeans equation; self-gravitating systems

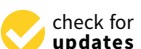



## 1. Introduction

The post-Newtonian approximations are solutions of Einstein's field equations in successive powers of the ratio $v/c$ where $v$ is a typical speed of the system and $c$ the light speed. It was proposed by Einstein, Infeld and Hoffmann [1] in 1938. The first post-Newtonian hydrodynamic equations for an Eulerian fluid were derived by Chandrasekhar [2] while the corresponding second post-Newtonian hydrodynamic equations were proposed by Chandrasekhar and Nutku [3].

The kinetic counterpart of the first post-Newtonian approximation was investigated in the works [4,5], where a collisionless Boltzmann equation was derived. Recently, the corresponding Boltzmann equation in the second post-Newtonian approximation was determined (see [6,7]).

The momentum density hydrodynamic equation for stationary self-gravitating systems, which follows from the collisionless Boltzmann equation, is denominated in astrophysics as the Jeans equation (see, e.g., [8] and the references therein). The Jeans equation is a differential equation for the momentum density of stationary self-gravitating systems which is coupled with the Poisson equation for the Newtonian gravitational potential.

The influence of the post-Newtonian approximation in the Jeans instability of self-gravitating systems was studied in the works [9–11].

The aim of this paper is to determine the first post-Newtonian approximation of the momentum density hydrodynamic equation for stationary spherically symmetrical self-gravitating systems, which corresponds to the first post-Newtonian approximation to the Jeans equation.

The post-Newtonian Jeans equation is coupled with the three Poisson equations from the post-Newtonian theory. One of them refers to the Poisson equation for the Newtonian gravitational potential, while the two others correspond to a scalar and a vector gravitational potential. The Poisson equations for the gravitational potentials are linked with the post-Newtonian approximations of the energy-momentum tensor components. Here, the energy-momentum tensor components are determined from the post-Newtonian equilibrium distribution function [7] which corresponds to the relativistic Maxwell–Jüttner equilibrium distribution function [7].

As an application of the theory developed here, the system of equations consisting of the stationary spherically symmetrical post-Newtonian Jeans equation and Poisson equations is solved numerically for the determination of the effect of a central massive black hole on the velocity dispersion profile of the host galaxy and the corresponding influence of the post-Newtonian corrections in the solution of this problem.

The work is outlined as follows: In Section 2, we introduce the post-Newtonian Poisson equations and calculate the energy-momentum tensor components from the post-Newtonian Maxwell–Jüttner equilibrium distribution function. The determination of the post-Newtonian collisionless Boltzmann equation in spherical coordinates is the subject of Section 3. The post-Newtonian Jeans equation for stationary self-gravitating systems is derived in Section 4 from the Boltzmann equation in spherical coordinates. The subject of Section 4 is the determination of the effect of a central massive black hole on the velocity dispersion profile of a host galaxy from the system of equations comprised by the post-Newtonian Jeans and Poisson equations. In the last section, the conclusions of the work are stated.

## 2. Poisson Equations

In the first post-Newtonian approximation, the components of the metric tensor are given by [12]

$$g_{00} = 1 + \frac{2\phi}{c^2} + \frac{2}{c^4}\left(\phi^2 + \psi\right), \qquad g_{0i} = -\frac{\xi_i}{c^3}, \qquad g_{ij} = -\left(1 - \frac{2\phi}{c^2}\right)\delta_{ij}, \tag{1}$$

where the scalar gravitational potentials $\phi$, $\psi$ and the vector gravitational potential $\xi_i$ satisfy the Poisson equations

$$\nabla^2\phi = \frac{4\pi G}{c^2}\overset{0}{T}{}^{00}, \qquad \nabla^2\psi = 4\pi G\left(\overset{2}{T}{}^{00} + \overset{2}{T}{}^{ii}\right) + \frac{\partial^2\phi}{\partial t^2}, \qquad \nabla^2\xi^i = \frac{16\pi G}{c}\overset{1}{T}{}^{0i}. \tag{2}$$

Above, $G$ denotes the universal gravitational constant. In the post-Newtonian theory, the energy-momentum tensor is split in orders of $(v/c)^n$ denoted by $\overset{n}{T}{}^{\mu\nu}$.

Within the framework of kinetic theory of gases, the phase space spanned by the spatial coordinates $\mathbf{x}$ and momentum $\mathbf{p}$ of the particles is characterized by the one-particle distribution function $f(\mathbf{x}, \mathbf{p}, t)$ and the energy-momentum tensor is written as [6,7]

$$T^{\mu\nu} = m^4c\int u^\mu u^\nu f\frac{\sqrt{-g}\,d^3u}{u_0}. \tag{3}$$

Above, $m$ is the fluid particle rest mass, $u^\mu = p^\mu/m$ the fluid particle four-velocity and $\sqrt{-g}\,d^3u/u_0$ the invariant integration element.

In the first post-Newtonian approximation the components of the four-velocity read [12]

$$u^0 = c\left[1 + \frac{1}{c^2}\left(\frac{v^2}{2} - \phi\right)\right], \qquad u^i = \frac{u^0 v_i}{c}, \tag{4}$$

where $v_i$ denotes the three-velocity of the fluid particle.

For a relativistic gas at equilibrium, the one-particle distribution function is given by the Maxwell–Jüttner distribution function (see, e.g., [7]). For a stationary system, the first post-Newtonian approximation of the one-particle distribution function is given by [7]

$$f = f_0\left\{1 - \frac{15kT}{8mc^2} - \frac{m}{kTc^2}\left[\frac{3v^4}{8} - 2\phi v^2\right]\right\}$$

$$= \frac{ne^{-\frac{mv^2}{2kT}}}{(2\pi mkT)^{\frac{3}{2}}}\left\{1 - \frac{15kT}{8mc^2} - \frac{m}{kTc^2}\left[\frac{3v^4}{8} - 2\phi v^2\right]\right\}. \tag{5}$$

Here, $f_0$ is the Maxwellian distribution function, $k$ the Boltzmann constant, $n$ the particle number density and $T$ the absolute temperature of the self-gravitating system.

In order to determine the components of the energy-momentum tensor from (3), we have to know the first post-Newtonian approximation of the invariant integration element $\sqrt{-g}\, d^3u/u_0$ whose expression is [7]

$$\frac{\sqrt{-g}\, d^3u}{u_0} = \left\{1 + \frac{1}{c^2}\left[2v^2 - 6\phi\right]\right\}\frac{d^3v}{c}. \tag{6}$$

The components of the energy-momentum tensor can be evaluated through the insertion of (4)–(6) into its definition (3) and integration of the resulting equations, yielding

$$\overset{0}{T}{}^{00} + \overset{2}{T}{}^{00} = \rho c^2 \left\{1 + \frac{1}{c^2}\left[3\langle v^2\rangle - 8\phi - \frac{15kT}{8m} - \frac{m}{kT}\left(\frac{3}{8}\langle v^4\rangle - 2\phi\langle v^2\rangle\right)\right]\right\}, \tag{7}$$

$$\overset{2}{T}{}^{11} = \rho\langle(v^1)^2\rangle, \quad \overset{2}{T}{}^{22} = \rho\langle(v^2)^2\rangle, \quad \overset{2}{T}{}^{33} = \rho\langle(v^3)^2\rangle, \quad \overset{2}{T}{}^{ij} = 0 \ (i \neq j), \quad \overset{1}{T}{}^{0i} = 0. \tag{8}$$

Here, we have introduced the mean values of the Maxwellian distribution function

$$\rho = \int m^4 f_0 d^3v, \qquad \rho\langle\chi\rangle = \int m^4 \chi f_0 d^3v, \tag{9}$$

where $\rho = mn$ is the fluid mass density and $\chi$ an arbitrary function of the particle velocities.

The Poisson Equation (2) by considering the post-Newtonian components of the energy-momentum tensor, (7) and (8) become

$$\nabla^2\phi = 4\pi G\rho, \qquad \nabla^2\xi^i = 0, \tag{10}$$

$$\nabla^2\psi = 4\pi G\rho\left[4\langle v^2\rangle - 8\phi - \frac{15kT}{8m} - \frac{m}{kT}\left(\frac{3}{8}\langle v^4\rangle - 2\phi\langle v^2\rangle\right)\right] + \frac{\partial^2\phi}{\partial t^2}. \tag{11}$$

We note that the component of the energy-momentum tensor $\overset{1}{T}{}^{0i}$ vanishes, since the Maxwellian distribution function is even in **v**. This implies that the Poisson equation for the gravitational potential $\vec{\xi}$ also vanishes and we can consider it as a Laplacian vector field where $\nabla \cdot \vec{\xi} = 0$ and $\nabla \times \vec{\xi} = 0$, i.e., since the former equation indicates that $\vec{\xi}$ is an incompressible field, it can be taken also as an irrotational field.

### 3. Post-Newtonian Boltzmann Equation

The space-time evolution of the one-particle distribution function $f(\mathbf{x}, \mathbf{v}, t)$ in the phase space spanned by the spatial coordinates **x** and fluid particle velocity **v** is governed by the Boltzmann equation, and the first post-Newtonian approximation of the Boltzmann equation for collisionless systems read [5,7]

$$\left[\frac{\partial f}{\partial t} + v^i\frac{\partial f}{\partial x^i}\right]\left[1 + \frac{1}{c^2}\left(\frac{v^2}{2} - \phi\right)\right] - \frac{\partial\phi}{\partial x^i}\frac{\partial f}{\partial v^i} + \frac{1}{c^2}\left[4v^iv^j\frac{\partial\phi}{\partial x^j} + 3v^i\frac{\partial\phi}{\partial t}\right.$$
$$\left. - \left(\frac{3v^2}{2} + 3\phi\right)\frac{\partial\phi}{\partial x^i} - \frac{\partial\psi}{\partial x^i} - \frac{\partial\xi_i}{\partial t} - v^j\left(\frac{\partial\xi_i}{\partial x^j} - \frac{\partial\xi_j}{\partial x^i}\right)\right]\frac{\partial f}{\partial v^i} = 0. \tag{12}$$

Without the relativistic corrections, the above equation reduces to the Boltzmann equation for the Newtonian gravitational potential $\phi$:

$$\frac{\partial f}{\partial t} + v^i\frac{\partial f}{\partial x^i} - \frac{\partial\phi}{\partial x^i}\frac{\partial f}{\partial v^i} = 0. \tag{13}$$

The post-Newtonian Boltzmann Equation (12) in spherical coordinates is obtained by using the relations of the Cartesian coordinates $(x^1, x^2, x^3)$ and spherical coordinates $(r, \theta, \varphi)$ together with the relations of the velocities $(v^1, v^2, v^3)$ in Cartesian coordinates

and the corresponding velocities in spherical coordinates $(v_r = \dot{r}, v_\theta = r\dot{\theta}, v_\varphi = r\sin\theta\dot{\varphi})$, namely

$$x^1 = r\sin\theta\cos\varphi, \quad x^2 = r\sin\theta\sin\varphi, \quad x^3 = r\cos\theta, \tag{14}$$

$$v^1 = v_r\sin\theta\cos\varphi + v_\theta\cos\theta\cos\varphi - v_\varphi\sin\varphi, \tag{15}$$

$$v^2 = v_r\sin\theta\sin\varphi + v_\theta\cos\theta\sin\varphi + v_\varphi\cos\varphi, \quad v^3 = v_r\cos\theta - v_\theta\sin\theta. \tag{16}$$

The material time derivative of the distribution function $f = f(t, r, \theta, \varphi, v_r, v_\theta, v_\varphi)$ in spherical coordinates reads

$$
\frac{df}{dt} = \frac{\partial f}{\partial t} + v^i\frac{\partial f}{\partial x^i} = \frac{\partial f}{\partial t} + \frac{\partial f}{\partial r}\dot{r} + \frac{\partial f}{\partial \theta}\dot{\theta} + \frac{\partial f}{\partial \varphi}\dot{\varphi} + \frac{\partial f}{\partial v_r}\dot{v}_r + \frac{\partial f}{\partial v_\theta}\dot{v}_\theta + \frac{\partial f}{\partial v_\varphi}\dot{v}_\varphi = \frac{\partial f}{\partial t}
$$
$$
+ v_r\frac{\partial f}{\partial r} + \frac{v_\theta}{r}\frac{\partial f}{\partial \theta} + \frac{v_\varphi}{r\sin\theta}\frac{\partial f}{\partial \varphi} + \left(\frac{v_\theta^2 + v_\varphi^2}{r}\right)\frac{\partial f}{\partial v_r} + \left(\frac{v_\varphi^2\cotan\theta}{r} - \frac{v_r v_\theta}{r}\right)\frac{\partial f}{\partial v_\theta}
$$
$$
- \left(\frac{v_\theta v_\varphi\cotan\theta}{r} + \frac{v_r v_\varphi}{r}\right)\frac{\partial f}{\partial v_\varphi}, \tag{17}
$$

so that the post-Newtonian Boltzmann Equation (12) in spherical coordinates becomes

$$
\left(1 + \frac{v^2}{2c^2} - \frac{\phi}{c^2}\right)\left[\frac{\partial f}{\partial t} + v_r\frac{\partial f}{\partial r} + \frac{v_\theta}{r}\frac{\partial f}{\partial \theta} + \frac{v_\varphi}{r\sin\theta}\frac{\partial f}{\partial \varphi} + \left(\frac{v_\theta^2 + v_\varphi^2}{r}\right)\frac{\partial f}{\partial v_r} + \left(\frac{v_\varphi^2\cotan\theta}{r}\right.\right.
$$
$$
\left.- \frac{v_r v_\theta}{r}\right)\frac{\partial f}{\partial v_\theta} - \left(\frac{v_\theta v_\varphi\cotan\theta}{r} + \frac{v_r v_\varphi}{r}\right)\frac{\partial f}{\partial v_\varphi}\Bigg] - \left(1 + \frac{3v^2}{2c^2} + \frac{3\phi}{c^2}\right)\left(\frac{\partial f}{\partial v_r}\frac{\partial \phi}{\partial r}\right.
$$
$$
+ \frac{1}{r}\frac{\partial f}{\partial v_\theta}\frac{\partial \phi}{\partial \theta} + \frac{1}{r\sin\theta}\frac{\partial f}{\partial v_\varphi}\frac{\partial \phi}{\partial \varphi}\right) + \frac{1}{c^2}\left\{3\left(v_r\frac{\partial f}{\partial v_r} + v_\theta\frac{\partial f}{\partial v_\theta} + v_\varphi\frac{\partial f}{\partial v_\varphi}\right)\frac{\partial \phi}{\partial t}\right.
$$
$$
+ 4\left(v_r\frac{\partial f}{\partial v_r} + v_\theta\frac{\partial f}{\partial v_\theta} + v_\varphi\frac{\partial f}{\partial v_\varphi}\right)\left(v_r\frac{\partial \phi}{\partial r} + \frac{v_\theta}{r}\frac{\partial \phi}{\partial \theta} + \frac{v_\varphi}{r\sin\theta}\frac{\partial \phi}{\partial \varphi}\right)
$$
$$
- \left(\frac{\partial f}{\partial v_r}\frac{\partial \psi}{\partial r} + \frac{1}{r}\frac{\partial f}{\partial v_\theta}\frac{\partial \psi}{\partial \theta} + \frac{1}{r\sin\theta}\frac{\partial f}{\partial v_\varphi}\frac{\partial \psi}{\partial \varphi}\right) - \frac{\partial f}{\partial v_r}\frac{\partial \xi_r}{\partial t} - \frac{\partial f}{\partial v_\theta}\frac{\partial \xi_\theta}{\partial t} - \frac{\partial f}{\partial v_\varphi}\frac{\partial \xi_\varphi}{\partial t}\right\} = 0. \tag{18}
$$

Here, we have considered that the gravitational potential $\xi_i$ is a Laplacian vector field.

For a stationary and spherically symmetrical self-gravitating system, the distribution function and the gravitational potentials do not depend on the angles $\theta$ and $\varphi$ but only on the radial coordinate $r$, and the post-Newtonian Boltzmann Equation (18) reduces to

$$
\left(1 + \frac{v^2}{2c^2} - \frac{\phi}{c^2}\right)\left[v_r\frac{df}{dr} + \left(\frac{v_\theta^2 + v_\varphi^2}{r}\right)\frac{\partial f}{\partial v_r} + \left(\frac{v_\varphi^2\cotan\theta}{r} - \frac{v_r v_\theta}{r}\right)\frac{\partial f}{\partial v_\theta}\right.
$$
$$
\left.- \left(\frac{v_\theta v_\varphi\cotan\theta}{r} + \frac{v_r v_\varphi}{r}\right)\frac{\partial f}{\partial v_\varphi}\right] - \left(1 + \frac{3v^2}{2c^2} + \frac{3\phi}{c^2}\right)\left(\frac{\partial f}{\partial v_r}\frac{d\phi}{dr}\right)
$$
$$
+ \frac{1}{c^2}\left\{4v_r\left(v_r\frac{\partial f}{\partial v_r} + v_\theta\frac{\partial f}{\partial v_\theta} + v_\varphi\frac{\partial f}{\partial v_\varphi}\right)\frac{d\phi}{dr} - \frac{\partial f}{\partial v_r}\frac{d\psi}{dr}\right\} = 0. \tag{19}
$$

## 4. Post-Newtonian Jeans Equation

The radial component of the momentum density equation results from the multiplication of the stationary post-Newtonian Boltzmann equation in spherical coordinates (19) by the radial component of the four-velocity $m^4 v_r u^0/c$, the subsequent integration of

the resulting equation and by considering the invariant integration element (6). Hence, it follows

$$\frac{d}{dr}\left\{\rho\left[\langle v_r^2\rangle\left(1-\frac{8}{c^2}\phi-\frac{15}{8c^2}\frac{kT}{m}\right)+\frac{1}{c^2}\left(3+\frac{2m\phi}{kT}\right)\langle v_r^2 v^2\rangle-\frac{3m}{8kTc^2}\langle v_r^2 v^4\rangle\right]\right\}$$

$$-\frac{\rho}{r}\left\{\left(1-\frac{8}{c^2}\phi-\frac{15}{8c^2}\frac{kT}{m}\right)\left(\langle v_\theta^2\rangle+\langle v_\varphi^2\rangle-2\langle v_r^2\rangle\right)+\frac{1}{c^2}\left(3+\frac{2m\phi}{kT}\right)\left[\langle v_\theta^2 v^2\rangle+\langle v_\varphi^2 v^2\rangle\right.\right.$$

$$\left.\left.-2\langle v_r^2 v^2\rangle\right]-\frac{3m}{8kTc^2}\left[\langle v_\theta^2 v^4\rangle+\langle v_\varphi^2 v^4\rangle-2\langle v_r^2 v^4\rangle\right]\right\}+\frac{\rho}{r}\cotan\theta\left\{\left(1-\frac{8}{c^2}\phi\right.\right.$$

$$\left.-\frac{15}{8c^2}\frac{kT}{m}\right)\langle v_r v_\theta\rangle+\frac{1}{c^2}\left(3+\frac{2m\phi}{kT}\right)\langle v_r v_\theta v^2\rangle-\frac{3m}{8kTc^2}\langle v_r v_\theta v^4\rangle\right\}+\rho\frac{d\phi}{dr}\left[1-\frac{4}{c^2}\phi\right.$$

$$\left.+\frac{4}{c^2}\langle v^2\rangle-\frac{4}{c^2}\langle v_r^2\rangle-\frac{15}{8c^2}\frac{kT}{m}-\frac{m}{kTc^2}\left(\frac{3}{8}\langle v^4\rangle-2\phi\langle v^2\rangle\right)\right]+\frac{\rho}{c^2}\frac{d\psi}{dr}=0. \quad (20)$$

We can express the mean values of the particle velocities which appear in the radial component of the moment density Equation (20) in terms of $\langle v_r^2\rangle$, $\langle v_\theta^2\rangle$ and $\langle v_\varphi^2\rangle$. Indeed, by introducing the Maxwellian distribution function $f_0$ and integration of the resulting equations, we have

$$\langle v^2\rangle=3\frac{kT}{m},\quad \langle v^4\rangle=15\left(\frac{kT}{m}\right)^2,\quad \langle v^2 v_r^2\rangle=5\frac{kT}{m}\langle v_r^2\rangle,\quad \langle v^4 v_r^2\rangle=35\left(\frac{kT}{m}\right)^2\langle v_r^2\rangle, \quad (21)$$

$$\langle v^2 v_\theta^2\rangle=5\frac{kT}{m}\langle v_\theta^2\rangle,\quad \langle v^4 v_\theta^2\rangle=35\left(\frac{kT}{m}\right)^2\langle v_\theta^2\rangle,\quad \langle v^2 v_\varphi^2\rangle=5\frac{kT}{m}\langle v_\varphi^2\rangle, \quad (22)$$

$$\langle v^4 v_\varphi^2\rangle=35\left(\frac{kT}{m}\right)^2\langle v_\varphi^2\rangle,\quad \langle v_r v_\theta\rangle=\langle v_r v_\theta v^2\rangle=\langle v_r v_\theta v^4\rangle=\langle v_r v_\theta v_\varphi^2\rangle=0. \quad (23)$$

Note from the above equations that the odd moments in the components vanish.

The multiplication of the stationary post-Newtonian Boltzmann Equation (20) by $m^4 v_\theta u^0/c$ and the integration of the resulting equation by considering the invariant integration element (6) leads to

$$\frac{\rho}{r}\cotan\theta\left(\langle v_\theta^2\rangle-\langle v_\phi^2\rangle\right)\left\{\left(1-\frac{8}{c^2}\phi-\frac{15}{8c^2}\frac{kT}{m}\right)+\frac{5kT}{mc^2}\left(3+\frac{2m\phi}{kT}\right)-\frac{105kT}{8mc^2}\right\}=0, \quad (24)$$

by taking into account (21)–(23). Here, we can infer that $\langle v_\theta^2\rangle=\langle v_\phi^2\rangle$. For the component $m^4 v_\phi u^0/c$, the stationary post-Newtonian Boltzmann equation furnishes no new condition.

The radial component of the momentum density Equation (20) by using the relationships (21)–(23) reduces to the post-Newtonian Jeans equation

$$\frac{d}{dr}\left[\rho\langle v_r^2\rangle\left(1+\frac{2\phi}{c^2}\right)\right]+2\rho\frac{\langle v_r^2\rangle\beta}{r}\left(1+\frac{2\phi}{c^2}\right)+\rho\frac{d\phi}{dr}\left[1+\frac{2\phi}{c^2}+\frac{\langle v_r^2\rangle(1-6\beta)}{2c^2}\right]$$

$$+\frac{\rho}{c^2}\frac{d\psi}{dr}=0. \quad (25)$$

Above, we consider that $\langle v_\theta^2\rangle=\langle v_\varphi^2\rangle$, introduce the velocity anisotropy parameter $\beta=1-\langle v_\theta^2\rangle/\langle v_r^2\rangle$ and use the relation

$$\frac{kT}{m}=\frac{\langle v_r^2\rangle}{3}=\frac{(3-2\beta)\langle v_r^2\rangle}{3}. \quad (26)$$

The velocity anisotropy parameter $\beta$ specifies the degree of radial anisotropy of the system. For perfectly radial orbits $\langle v_\theta^2 \rangle = 0$ and $\beta = 1$, while for circular orbits $\langle v_r^2 \rangle = 0$ and $\beta = -\infty$. A radial bias is when $\beta > 0$ and a tangential bias is when $\beta < 0$.

Equation (25) can be rewritten as

$$\frac{d\rho\langle v_r^2 \rangle}{dr}\left(1 + \frac{2\phi}{c^2}\right) + 2\rho\frac{\langle v_r^2 \rangle\beta}{r}\left(1 + \frac{2\phi}{c^2}\right) + \rho\frac{d\phi}{dr}\left[1 + \frac{2\phi}{c^2} + \frac{\langle v_r^2 \rangle(5 - 6\beta)}{2c^2}\right]$$
$$+ \frac{\rho}{c^2}\frac{d\psi}{dr} = 0, \qquad (27)$$

which through the multiplication by $\left(1 - \frac{2\phi}{c^2}\right)$ and retaining terms up to the order $1/c^2$, the post-Newtonian Jeans equation reduces to

$$\frac{d\rho\langle v_r^2 \rangle}{dr} + 2\rho\frac{\beta\langle v_r^2 \rangle}{r} + \rho\frac{d\phi}{dr}\left[1 + \frac{\langle v_r^2 \rangle(5 - 6\beta)}{2c^2}\right] + \frac{\rho}{c^2}\frac{d\psi}{dr} = 0. \qquad (28)$$

Without the $1/c^2$ contributions, this equation becomes the Newtonian Jeans equation for the radial component of the momentum density [8], namely

$$\frac{d\rho\langle v_r^2 \rangle}{dr} + 2\rho\frac{\beta\langle v_r^2 \rangle}{r} + \rho\frac{d\phi}{dr} = 0. \qquad (29)$$

The post-Newtonian Jeans Equation (28) is coupled with the stationary Poisson Equations (10) and (11) which in spherical coordinates become

$$\frac{1}{r^2}\frac{d}{dr}\left(r^2\frac{d\phi}{dr}\right) = 4\pi G\rho, \qquad (30)$$

$$\frac{1}{r^2}\frac{d}{dr}\left(r^2\frac{d\psi}{dr}\right) = 4\pi G\rho\left[\frac{3\langle v_r^2 \rangle(3 - 2\beta)}{2} - 2\phi\right]. \qquad (31)$$

From the system of differential Equations (28), (30) and (31), one may determine the mean value of the radial velocity square $\langle v_r^2 \rangle$ as function of the radial distance, once the dependence of the mass density $\rho$ on the radial distance $r$ is specified.

## 5. Velocity Dispersion Profile

In this section, we shall search for a solution of the system of differential Equations (28), (30) and (31) which corresponds to the effect of a central massive black hole on the velocity dispersion profile of the host galaxy [8].

We begin by introducing the dimensionless variables

$$\rho_* = \frac{\rho}{\rho_0}, \qquad \sigma_r^2 = \frac{m}{kT}\langle v_r^2 \rangle, \qquad \phi_* = \frac{m}{kT}\phi, \qquad \psi_* = \frac{m^2}{k^2T^2}\psi, \qquad (32)$$

$$r_* = \frac{\sqrt{4\pi G\rho_0}}{kT/m}r = k_J r, \qquad \zeta = \frac{mc^2}{kT}. \qquad (33)$$

Above, $\rho_0$ is a reference mass density of the self-gravitating system and $k_J$ the Jeans wave number. Furthermore, the relativistic parameter $\zeta$ depends on the absolute temperature of the system and is given by the ratio of the rest energy of a particle $mc^2$ and the thermal energy of the system $kT$.

The system of differential Equations (28), (30) and (31) in terms of the above dimensionless variables reads

$$\frac{d\rho_* \sigma_r^2}{dr_*} + 2\rho_* \frac{\beta \sigma_r^2}{r_*} + \rho_* \frac{d\phi_*}{dr_*}\left[1 + \frac{\sigma_r^2(5 - 6\beta)}{2\zeta}\right] + \frac{\rho_*}{\zeta}\frac{d\psi_*}{dr_*} = 0, \tag{34}$$

$$\frac{1}{r_*^2}\frac{d}{dr_*}\left(r_*^2 \frac{d\phi_*}{dr_*}\right) = \rho_*, \tag{35}$$

$$\frac{1}{r_*^2}\frac{d}{dr_*}\left(r_*^2 \frac{d\psi_*}{dr_*}\right) = \rho_*\left[\frac{3\sigma_r^2(3 - 2\beta)}{2} - 2\phi_*\right]. \tag{36}$$

In the Newtonian limiting case, the above system of differential equations reduce to

$$\frac{d\rho_* \sigma_r^2}{dr_*} + 2\rho_* \frac{\beta \sigma_r^2}{r_*} + \rho_* \frac{d\phi_*}{dr_*} = 0, \qquad \frac{1}{r_*^2}\frac{d}{dr_*}\left(r_*^2 \frac{d\phi_*}{dr_*}\right) = \rho_*. \tag{37}$$

If we know the mass density profile $\rho_*$, the relativistic parameter $\zeta$ and the velocity anisotropy parameter $\beta$, the system of differential Equations (34)–(36) can be solved numerically.

Here, we follow the book of Binney and Tremaine [8] and assume that the galaxy has a constant mass-to-light ratio while the Newtonian gravitational potential and the corresponding mass density are given by the Hernquist model of scale-length $a$. In terms of dimensionless quantities, the dimensionless mass density and the dimensionless Newtonian gravitational potential for the Hernquist model read [8]

$$\rho_* = \frac{2}{r_*(r_* + 1)^3}, \qquad \phi_* = -\frac{1}{r_* + 1} - \frac{\mu}{r_*}. \tag{38}$$

Here, $\mu = M_\bullet / M_g$ is the ratio of the black hole mass $M_\bullet$ and the galaxy mass $M_g$ which is associated with the mass density $\rho_0$, i.e., the mass of the self-gravitating system. Furthermore, the scale-length $a$ is identified with the inverse of Jeans wave number $a = 1/k_J$.

Note that the mass density and Newtonian gravitational potential in the Hernquist model (38) satisfy the Newtonian Poisson Equation (35).

The system of coupled differential Equations (34) and (36) was solved numerically for the dimensionless radial velocity dispersion $\sigma_r$ and gravitational potential $\psi_*$ where it was assumed the boundary conditions $\sigma_r(3) = 0.1$, $\psi_*(3) = 0$ and $d\psi_*(3)/dr_* = 0$ and different values of the relativistic parameter $\zeta$. Moreover, the values for the ratio of the black hole and galaxy masses $\mu = M_\bullet / M_g$ and the velocity anisotropy parameter $\beta = 1 - \langle v_\theta^2 \rangle / \langle v_r^2 \rangle$ adopted here are similar to the ones given in the book by Binney and Tremaine [8]. The values for the velocity anisotropy parameter adopted are: $\beta = 0.1$ corresponding to a radial bias and $\beta = -0.1$ to a tangential bias. The value for the ratio of the black hole and galaxy masses was fixed as $\mu = 0.004$.

Here, we introduce the dimensionless root mean square velocity dispersion $v_{rms}$ which is defined in terms of the dimensionless radial velocity dispersion $\sigma_r$ and velocity anisotropy parameter $\beta$ by

$$v_{rsm} = \sqrt{\frac{\langle v_r^2 \rangle + \langle v_\theta^2 \rangle + \langle v_\varphi^2 \rangle}{kT/m}} = \sqrt{3 - 2\beta}\,\sigma_r. \tag{39}$$

Figure 1 shows the behavior of the dimensionless root mean square velocity dispersion $v_{rms}$ as function of the dimensionless radial distance $r_*$ for a radial bias $\beta = 0.1$, ratio of the black hole and galaxy masses $\mu = 0.004$ and different values of the relativistic parameter $\zeta = mc^2/kT = 1.0 \times 10^3; 1.5 \times 10^3; 3.0 \times 10^3$. The post-Newtonian solutions are represented by dashed lines and the Newtonian solution—which follows from (37) by taking into account the same boundary conditions—by a solid line . From this figure, we may infer that the black hole has an influence on the dimensionless root mean square

velocity dispersion, since it increases at small radii. This is a consequence that the velocities of the stars increase due to the deep potential well of the black hole. The post-Newtonian corrections depend on the relativistic parameter $\zeta$—which decreases with the increase in the absolute temperature of the system—and on the post-Newtonian scalar potential $\psi_*$.

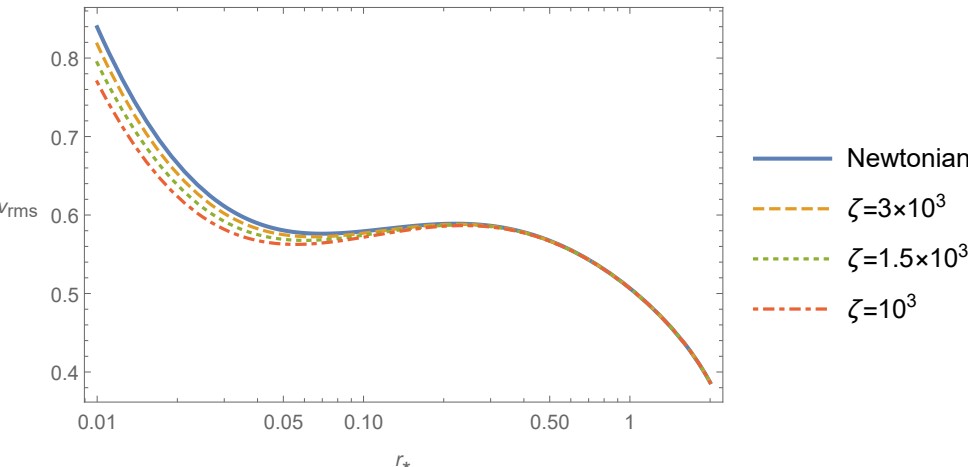

**Figure 1.** $v_{rms}$ as function of $r_*$ for $\beta = 0.1$, $\mu = 0.004$. Solid line corresponds to Newtonian solution and dashed lines to post-Newtonian solutions for $\zeta = 1.0 \times 10^3$; $1.5 \times 10^3$; $3.0 \times 10^3$.

Figure 2 displays the ratio $\psi_*/\zeta$ as function of $r_*$ showing that the post-Newtonian scalar gravitational potential is a positive quantity that increases with the decrease in the radial distance. One can infer from the behavior of $\psi_*/\zeta$ that it is responsible for the attenuation of the increase in the root mean square velocity dispersion when the radial distance decreases as shown in Figure 1.

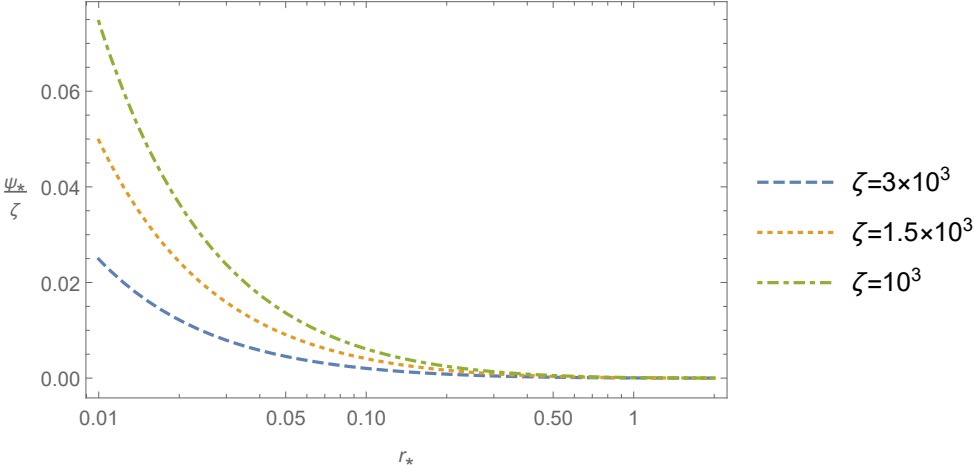

**Figure 2.** $\psi_*/\zeta$ as function of $r_*$ for $\beta = 0.1$, $\mu = 0.004$ and $\zeta = 1.0 \times 10^3$; $1.5 \times 10^3$; $3.0 \times 10^3$.

In Figure 3, the dimensionless root mean square velocity dispersion $v_{rms}$ is plotted as function of the dimensionless radial distance $r_*$ for a tangential bias $\beta = -0.1$ and the same previous values for $\mu = 0.04$ and $\zeta = 1.0 \times 10^3, 1.5 \times 10^3, 3.0 \times 10^3$. From this figure, one may infer the same behavior of the curves as those in the preceding case, but here the values of the dimensionless root mean square velocity dispersion are smaller in comparison to the previous case.

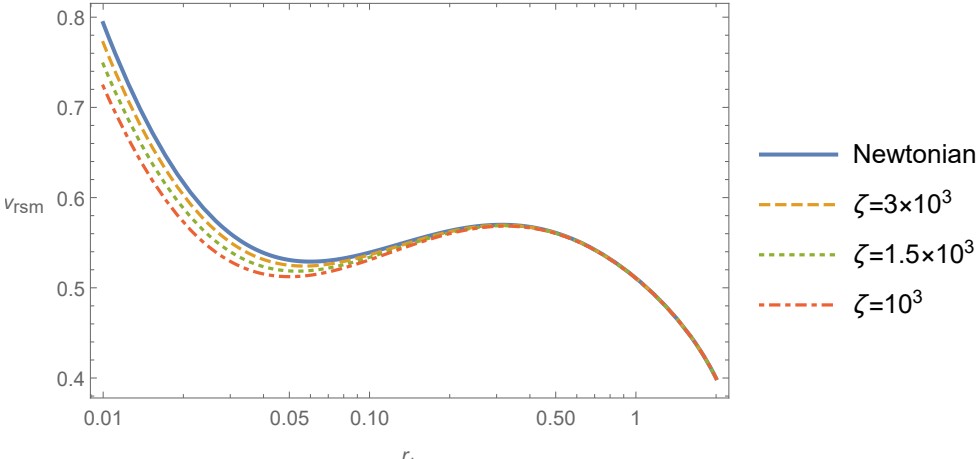

**Figure 3.** $v_{rms}$ as function of $r_*$ for $\beta = -0.1$, $\mu = 0.004$. Solid line corresponds to Newtonian solution and dashed lines to post-Newtonian solutions for $\zeta = 1.0 \times 10^3$; $1.5 \times 10^3$; $3.0 \times 10^3$.

## 6. Conclusions

In this work, we have determined the stationary radial component of the momentum density hydrodynamic equation from the post-Newtonian Boltzmann in spherical coordinates. It represents the post-Newtonian Jeans equation for stationary spherically symmetrical self-gravitating systems, which is coupled with the three Poisson equations of the first post-Newtonian theory. The Poisson equations are functions of the energy-momentum tensor components which were determined by the use of the post-Newtonian Maxwell–Jüttner equilibrium distribution function. The coupled system of differential equations comprised by the Jeans and Poisson equations were solved numerically for the analysis of the effect of a central massive black hole on the velocity dispersion profile of the host galaxy. For the Newtonian gravitational potential and corresponding mass density, it was supposed that they are given by the scale-length Hernquist model. It was shown that the post-Newtonian solution for the root mean square dispersion velocity has a less accentuated behavior by decreasing the radial coordinate than the one for the Newtonian solution. This behavior is due to the presence of the post-Newtonian scalar gravitational potential $\psi_*$ which is always positive.

**Funding:** This research received no external funding.

**Institutional Review Board Statement:** Not applicable.

**Informed Consent Statement:** Not applicable.

**Acknowledgments:** This work was supported by Conselho Nacional de Desenvolvimento Científico e Tecnológico (CNPq), grant No. 304054/2019-4.

**Conflicts of Interest:** The author declares no conflict of interest.

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
