# Peer review of "Post-Newtonian Jeans Equation for Stationary and Spherically Symmetrical Self-Gravitating Systems"

_universe, doi:10.3390/universe8030179_

Round 1

Reviewer 1 Report

This paper discusses the post-Newtonian Jeans equation for stationary self-gravitating systems. Specifically, the paper presents a derivation based on the post-Newtonian Boltzmann equation in spherical coordinates. The main results of the paper are applied to the problem of velocity dispersion profile in galaxies with central black-holes. The paper is clearly written and the discussion is timely and pertinent, I recommend its publication in the Universe. 

Author Response

The paper was revised with respect to the spell check. 

Reviewer 2 Report

%\documentclass[preprint,amsmath,amssymb]{revtex4-1}
\documentclass[preprint,amsmath,amssymb]{revtex4}
\usepackage{amsmath}

%%%
\newcommand{\Eqn}[1]{&\hspace{-0.2em}#1\hspace{-0.2em}&}
\newcommand{\rd}{{\rm d}}
\newcommand{\abs}[1]{\vert{#1}\vert}
%%%

\begin{document}
\tolerance=5000 \noindent
Manuscript ID: Universe-1571373 \\
Title: ``Post-Newtonian Jeans Equation for Stationary and
Spherically Symmetrical Self-Gravitating Systems"\\
Authors: Gilberto M. Kremer \\
Universe.
%%%%%

\vspace{5mm}

\noindent
%%%%%%%%
{\em ``The author have presented in the manuscript entitled
``Post-Newtonian Jeans Equation for Stationary and Spherically
Symmetrical Self-Gravitating Systems'' interesting mathematical
analysis by assuming several ansatzes and have discussed their
results based on the graphical plots. However, a few physical issues
are needed to be clarified which come immediately in my observations
as follows:\\
(1)What is physical basis for opting spherical symmetry? \\
(2)In Eq.\textbf{3} the term $d^{3}u$ appears. What does it
represent? You should mention it in
the manuscript.\\
(3)After Eq.\textbf{11} you have written ``since the Maxwellian
distribution function is even in $v$''. Can't it be odd or zero?
What will happen in that cases? \\
(4)Define Laplacian vector field also why you have considered the
the gravitational potential $\vec{\zeta}$ as Laplacian vector field?
What is the reason behind it?I request the author to mention about it in the manuscript. \\
(5)Why you have chosen the dimensionless variables in velocity
dispersion profile?\\
(6)After Eq.\textbf{38} you have written ``the ratio of the black
hole mass $M_{\bigodot}$ and the galaxy mass $M_{g}$ which is
associated with the mass density $\rho_{0}$''. How these two ratios
are related to the reference mass density of the self-gravitating
system? \\
(7)Please explain. Why do you plot the graphs considering the positive values of
$\zeta$ only? What will be scenario for $\zeta\leq 0$ ? Discuss them too.\\
(8)Please describe the physical significance of the velocity anisotropy
parameter $\beta$? What does $\beta\gtreqqless0$ represents? I also
request the author to discuss each case, separately and add it
in conclusion of your manuscript.\\
(9)Please relate your results with Dynamical instability of the charged expansion-free spherical collapse in f(R) gravity. 
This could help the author to understand and related the results with Post-Newtonian limits.\\

Based on the above points, I personally recommend - ``Major
Revision".
%%%%%%%%

\end{document}

Author Response

see pdf

Reviewer 3 Report

Post Newtonian corrections to the behavior of a spherically symmetric cloud are presented. The results are derived within the standard approach and look reasonable. But there is one important issue which should have been addressed in such a study. Namely, it is required to verify how do the post Newtonian corrections affect the Jeans instability in the considered system.

I recommend publication of the article upon a reply to the above remark 

Minor corrections the English language are also required, see, e.g. the last sentence of Abstract, where "is investigate" should read as "is investigated".

Author Response

I have added the paragraph in the introduction and the references:

The influence of the post-Newtonian  approximation in the Jeans instability of self-gravitating systems was studied in the works [8-10]

8. Nazari E.; Kazemi A.;  Roshan M.; Abbassi S. Post-Newtonian Jeans analysis. \emph{Ap. J.} {\bf2017}, {\emph839}, 839.
9. Noh H.;  Hwang J-C. 
Gravitomagnetic instabilities of relativistic magnetohydrodynamics. \emph{Ap. J.}, {\bf2021}, {\emph906}, 22.
10. Kremer G. M. Jeans instability from post-Newtonian Boltzmann equation. \emph{Eur. Phys. J. C} {\bf2021}, {\emph81}, 927

I corrected  the "is investigated" thank you.